# Analysis of influencing factors of no/low response to preoperative concurrent chemoradiotherapy in locally advanced rectal cancer

**Fengpeng Wu**[1], **Guiying Wang**[2]*, **Jun Wang**[1], **Chaoxi Zhou**[2], **Congrong Yang**[1], **Wenbo Niu**[2], **Jianfeng Zhang**[2], **Guanglin Wang**[2], **Yafan Yang**[2]

**1** Department of Radiation Oncology, Hebei Medical University Fourth Affiliated Hospital and Hebei Provincial Tumor Hospital, Shijiazhuang, China, **2** Department of Gastrointestinal Surgery, Hebei Medical University Fourth Affiliated Hospital and Hebei Provincial Tumor Hospital, Shijiazhuang, China

* wgywfp@163.com

## Abstract

The aim of this study is to investigate the influencing factors associated with no/low response to preoperative concurrent chemoradiotherapy (CCRT) for locally advanced rectal cancer (LARC) patients. A total of 79 patients were included in this prospective study. Fifteen factors that might affect the resistance to CCRT were included in this logistic regression analysis, these factors include the general clinical data of patients, the expression status of tumor stem cell marker CD44v6 and the volumetric imaging parameters of primary tumor lesions. We found that the no/low response status to preoperative CCRT was positively correlated with the real tumor volume (RTV), the total surface area of tumor (TSA), and CD44v6 expression, whereas negatively correlated with the tumor compactness (TC). According to the results of logistic regression analysis, two formulas that could predict whether or not no/low response to preoperative CCRT were established. The Area Under Curve (AUC) of the two formulas and those significant measurement data (RTV, TC, TSA) were 0.900, 0.858, 0.771, 0.754, 0.859, the sensitivity were 95.8%, 79.17%, 62.50%, 95.83%, 62.5%, the specificity were 70.9%, 74.55%, 83.64%,47.27%, 96.36%, the positive predictive values were 58.96%, 57.58%, 62.51%,44.23%, 88.23%, the negative predictive values were 97.48%, 89.13%, 83.64%, 96.29%, and 85.48%, respectively.

## Introduction

Preoperative concurrent chemoradiotherapy (CCRT), as an important part of the comprehensive treatment of locally advanced rectal cancer (LARC), has been widely applied in clinical practice with an effective rate of approximately 40–80% [1–4]. However, there are many patients who cannot benefit from this type of treatment. In addition, preoperative CCRT may result in pelvic connective tissue fibrosis, radiation enteritis, along with other side effects that can lead to postoperative complications such as anastomotic leakage, and defects in wound

**Funding:** This work was supported by the Science and Technology Department of Hebei Province ((Grant Number 17277750D), http://kjt.hebei.gov.cn/www/index.html. The funders had no role in study design, data collection and analysis, decision to publish, or preparation of the manuscript.

**Competing interests:** NO authors have competing interests.

healing. Therefore, it is of great significance to explore a method of selecting patients with no/low response to CCRT before treatment.

Cancer stem cells (CSCs) have a strong self-renewal ability, DNA repair ability, hypoxic survival ability, and high radiation resistance [5–9]. It was found that CD44 positive tumor cells have a series of stem cell characteristics, such as strong proliferation, invasion, anti-apoptosis, and stable passage [10–14]. Therefore, the researchers believe that CD44 can be used as an important marker of CSCs [15, 16].

Currently, accumulating clinical evidence is emerging, indicating that CD44 and its variant isoforms can predict a patient's resistance to radiation. Xiao et al. [17] found that the up-regulated expression of CD44 contributes to the cell cycle arrest of prostate cancer cells and the repair of DNA double-strand breaks (DSB), consequently enhancing the radiation resistance. Huh et al. screened the expression of 13 factors that may affect the efficacy of preoperative CCRT in 123 LARC patients and found that the up-regulation of CD44 expression in tumor tissues before treatment predicted poor tumor regression [18]. Sagawa K et al. [15] found that patients with nasopharyngeal carcinoma and high expression of CD44v6 had poor clinical response to CCRT, and analysis of its expression would be helpful in predicting the prognosis of patients.

In view of this, the study also included the expression of CD44v6 in primary tumor tissue. At the same time, the general clinical data (age, gender, clinical T stage, clinical N stage, carcinoembryonic antigen (CEA) and pathological type) and imaging volume parameters (tumor maximum longitudinal length (TML), tumor maximum transverse diameter (TMD), approximate tumor volume (ATV), real tumor volume (RTV), tumor surface area inner the intestine (TSAI), tumor surface area outside the intestine (TSAO), total surface area of tumor (TSA), and tumor compactness(TC)) of the primary tumor were included in the study. In order to find a method of predicting the resistance of patients with LARC to CCRT, we used the above 15 factors to analyze the influence of no/low response of 79 patients with LARC after receiving long-course preoperative CCRT at our institute.

## Materials and methods

### Patients

The ethics committee of the Fourth Hospital of Hebei Medical University approved this study (2014MEC067). All participants provided written informed consent for inclusion to the study. From May 2015 to August 2017, 79 LARC patients who received long-course preoperative CCRT in our hospital were enrolled. All participants' information was available to the study group during and after data collection. The eligibility criteria included histologically confirmed adenocarcinoma or mucinous carcinoma, a distance of less than 10cm between the inferior margin of the tumor and the anal verge, clinical T2N+ or clinical T3–4 classification, as well as no distant metastasis confirmed by enhanced abdomen-pelvic MRI and chest CT, availability of contrast-enhanced CT for three-dimensional radiotherapy positioning.

### Preoperative CCRT and surgery

All patients were treated with long-course preoperative CCRT. The total dose of radiotherapy was 50.4Gy and the single dose was 1.8Gy. The target volume delineation and field setup were completed with reference to the ICRU Report 83 and the academic writings of Lee N Y et al. [19]. The chemotherapy regimen was to take capecitabine (825mg /m2, bid) orally concurrently with irradiation. Total mesorectal excision (TME) surgery was performed by two surgeons with more than 15 years of experience at 8 to 10 weeks after CCRT, each of whom completed at least 100 rectal cancer surgeries per-year.

## The expression of CD44v6

Rectal primary tumor samples obtained pre-treatments by colonoscopy were embedded and sliced into 4-μm-thick sections. The immunohistochemical staining of CD44v6 was performed using the ABC method according to the manufacturer's instructions (Abcam). Breast cancer specimens were used to act as positive control samples, as earlier studies have described that the positive expression rate of CD44v6 in breast cancer tissue was 98.06% [20]. PBS substituted primary antibodies were used as negative controls. The results were evaluated by two experienced pathologists who were blind to patient data. The staining scores were performed as follows: no staining or less than 10% positive cells were defined as "-", 10–20% weakly to moderately positive cells were "+", 10–20% intensively positive cells or 20–50% weakly positive cells were "+ +", 20–50% positive cells with moderate to strong reactivity or greater than 50% positive cells were "+++" [21].

## Volumetric imaging parameters

Pelvic high-resolution MRIs were performed before two weeks of treatment. Referring to these MRI images, we obtained direct and indirect imaging parameters on the radiotherapy localization CT images using Pinnacle version 9.1 system. Direct imaging parameters can be measured and calculated directly from CT images, including the approximate tumor volume (ATV), the tumor maximum longitudinal length (TML) and the tumor maximum transverse diameter (TMD). Indirect image parameters are obtained through the contraction and enlargement function of Pinnacle software, as shown in Fig 1, including real tumor volume (RTV), tumor surface area inner the intestine(TSAI), tumor surface area outside the intestine (TSAO), total surface area (TSA) and tumor compactness (TC). It should be noted that TC is a secondary derivative parameter, which can be calculated from the following equation [22–24],

$$\text{tumor compactness} = \frac{\text{real tumor volume}}{\text{total surface area}^{1.5}}.$$

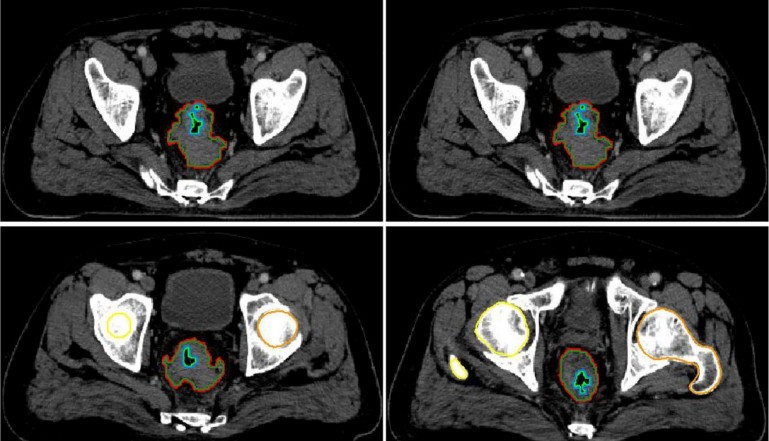

**Fig 1. Graphical representation of the method for obtaining indirect volumetric image parameters.** The red line encompasses the approximate tumor volume (ATV). The green line, was generated from ATV with a 3-dimensional universal contraction of 1mm length, and the tumor surface area (TSAO) was the area between the red line and the green line, encompasses the 1mm layer volume. The light green line encompasses the intestinal tube cavity (ITC), the area between the red line and light green line is the real tumor volume (RTV). The blue line was generated from the ITC with a 3-dimensional universal enlargement of 1mm length, and the tumor surface area inner the intestine (TSAI) was the area between the blue line and the light green line, encompasses the 1mm layer volume. Total surface area (TSA) was the sum value of TSAO and TSAI.

### Evaluation of pathologic response to preoperative CCRT

After 8 weeks of CCRT, we performed a preoperative MRI examination and compared with images before CCRT to evaluate the changes of the primary tumor, local positive lymph nodes and circumferential resection margin (CRM). Since the MRI results cannot fully reflect the tumor pathological regression status [25], especially its limitation in distinguishing residual tumor from surrounding fibrosis [26–28], we evaluated the rectal cancer regression grade (RCRG) of the enrolled patients by referring to the quantification standard of histologic regression of rectal cancer after irradiation of Wheeler JM et al. [29]. Within this scoring criterion, the regression of rectal tumors was classified into three levels: RCRG 1: Sterilization or only microscopic foci of adenocarcinoma remaining, with marked fibrosis; RCRG 2: Marked fibrosis but macroscopic disease present; RCRG 3: Little or no fibrosis, with abundant macroscopic disease. In our study, we defined RCRG 3 as no/low response.

### Statistical analysis

Statistical analysis was performed using SPSS software 17.0 (SPSS, Inc., Chicago, IL, USA) and MedCalc Version 16.2. The relationships between the factors and the no/low response status are analyzed by forward stepwise Logistic regression model. The Hosmer-Lemeshow test was used to evaluate the goodness of fit of logistic regression model. A nomogram was formulated based on the results of logistic regression analysis and by using the Empower Stats software of R, version 3.0(*http://www.empowerstats.com*). Receive operating characteristic (ROC) curve analysis was used to evaluate the prediction performance of the logistic regression model and the independent predictors. In our study, $P<0.05$ was considered statistically significant.

## Results

### Treatment characteristics and tumor response

In this study, all enrolled patients completed preoperative CCRT according to the pre-established neoadjuvant treatment regimen. MRI images taken one week before resection showed that the down-staging rates of tumors and lymph nodes were 53.16% (42 cases) and 55.70% (44 cases), respectively, and the CRM status of all patients were negative. According to TME standards, surgical resection was performed 8–10 weeks after CCRT. Out of all enrolled patients, 17 underwent abdominoperineal resection, 62 underwent low anterior resection, none of whom underwent Hartmann surgery. Of the 62 patients who underwent low anterior resection, 12 underwent temporary protective loop ileostomy. The proximal and distal margins of all specimens were negative. The RCRG status of enrolled patients has been reported in our previous study [30] as follows: 20 (25.32%) cases for RCRG 1, 35(44.30%) for RCRG 2, 24 (30.38%) for RCRG 3, resulting in 24 patients showing no/low response to preoperative CCRT. There were 5 cases with positive CRM, all of which were found in RCRG 3 group. From a total of 18 cases with Lymph-vascular space invasion, 3 were found in RCRG 2 group and 15 found in RCRG 3 group. The relationship between the general characteristics of patients and the response to preoperative CCRT is shown in Table 1.

### The expression of CD44v6

Of the 79 patients examined for CD44v6, 33 (41.77%) had"-" expression, 21 (26.58%) had"+" expression, 17 (21.52%) had"++" expression, 8 (10.13%) was "+++" expression. These results have been reported in our previous study [30].

**Table 1. Baseline demographics of all patients (N = 79).**

| Characteristic | Classification and delamination | Number | No/low response to preoperative CCRT(n = 24) | Obvious response preoperative CCRT(n = 55) |
|---|---|---|---|---|
| Age(years) | ≤40 | 5 | 2(8.33%) | 3(5.45%) |
| | 41–50 | 14 | 2(8.33%) | 12(21.82%) |
| | 51–60 | 33 | 11(45.83%) | 22(40.00%) |
| | 61–70 | 18 | 5(20.83%) | 13(23.64%) |
| | ≥71 | 9 | 4(16.67%) | 5(9.09%) |
| Gender | Male | 46 | 14(58.33%) | 32(58.18%) |
| | Female | 33 | 10(41.67%) | 23(41.82%) |
| Clinical T stage | 2 | 11 | 3(12.50%) | 8(14.55%) |
| | 3 | 54 | 16(66.67%) | 38(69.09%) |
| | 4 | 14 | 5(20.83%) | 9(16.36%) |
| Clinical N stage | 0 | 11 | 1(4.17%) | 10(18.18%) |
| | 1 | 39 | 8(33.33%) | 31(56.36%) |
| | 2 | 29 | 15(62.50%) | 14(25.45%) |
| CEA(ng/ml) | ≤5 | 39 | 11(45.83%) | 28(50.91%) |
| | >5 | 40 | 13(54.17%) | 27(49.09%) |
| Pathological pattern | Non-mucinous adenocarcinoma | 69 | 21(87.50%) | 48(87.27%) |
| | Mucinous adenocarcinoma | 10 | 3(12.50%) | 7(12.73%) |

CEA, Carcinoembryonic antigen.

## Correlation between parameters

Under the premise of no/low response to preoperative CCRT being a dependent variable, we selected age, gender, clinical T stage, clinical N stage, pathological pattern, serum CEA levels, CD44v6 expression, TML, TMD, ATV, RTV, TSAI, TSAO, TSA, along with TC as possible predictors on the basis of previous results [30], we stratified each factor according to its characteristics (Table 2), and provided its value in the database (S1 Data). Collinearity analysis for all variables were performed before logistic regression analysis, and the results show that the variance inflation factor (VIF)≥7, including RTV (8.124) and ATV (7.391), were thought to be highly correlated with at least one of the other factors in the regression model. Considering that both ATV and RTV are descriptions of tumor size, we incorporated one of the two variables in the collinear analysis respectively, and found that, with either ATV or RTV included, VIF of all variables in the analysis model were still less than 6 (Table 3).

## Multivariate analyses for no/low response status

According to the results of the previous step, we found strong collinearity between ATV and RTV. We then separated them into subsequent goodness of fit test and multivariate analysis models. The significance of the Hosmer-Leme show goodness of fit was 0.318(incl RTV) and 0.886 (incl ATV) respectively, indicating that the model has a good degree of fit (*P>0.05*). Multivariate analysis showed that, with deleting ATV from the model, TC was a negative predictor for no/low response status (*P = 0.018*), RTV and the CD44v6 expression were both positive predictors (*P = 0.001, P = 0.008*) (Table 4), Logistic regression model was $Z1 = sigmoid(X) = \frac{1}{1+e^{-4.419+1.003 \times X7+1.557 \times X11-1.004 \times X15}}$. As we excluded RTV from the model, the results showed that CD44v6 expression and TSA were positive predictive factors (*P = 0.006, P = 0.008*) (Table 4), Logistic regression model was $Z2 = sigmoid(X) = \frac{1}{1+e^{-6.547+0.864 \times X7+2.081 \times X14}}$. In the above models, Z1 and Z2 represent the probability of patients with no/low response to

**Table 2. The layering of all variable in regression model.**

| Variables | Label | The explanation of layering | | | | |
|---|---|---|---|---|---|---|
| Hierarchy | | 1 | 2 | 3 | 4 | 5 |
| Age(years) | X1 | ≤40 | 41–50 | 51–60 | 61–70 | ≥71 |
| Gender | X2 | Male | Female | | | |
| Clinical T stage | X3 | T2 | T3 | T4 | | |
| Clinical N stage | X4 | N0 | N1 | N2 | | |
| pathological pattern | X5 | Non-mucinous adenocarcinoma | Mucinous adenocarcinom | | | |
| CEA(ng/mL) | X6 | ≤5 | >5 | | | |
| CD44v6 | X7 | "-" | "+" | "++" | "+++" | |
| TML (cm) | X8 | ≤5 | >5&≤8 | >8&≤11 | >11 | |
| TMD (cm) | X9 | ≤3 | >3&≤5 | >5&≤7 | >7 | |
| ATV (cm$^3$) | X10 | ≤35 | >35&≤70 | >70&≤105 | >105&≤140 | >140 |
| RTV (cm$^3$) | X11 | ≤35 | >35&≤70 | >70&≤105 | >105&≤140 | >140 |
| TSAI (cm$^2$) | X12 | ≤2 | >2&≤4 | >4&≤6 | >6 | |
| TSAO (cm$^2$) | X13 | ≤10 | >10&≤20 | >20&≤30 | >30 | |
| TSA (cm$^2$) | X14 | ≤10 | >10&≤20 | >20&≤30 | >30 | |
| TC | X15 | ≤1.0 | >1.0&≤1.5 | >1.5&≤2 | >2&≤2.5 | >2.5 |

CEA, Carcinoembryonic antigen; TML, tumor maximum longitudinal length; TMD, tumor maximum transverse diameter; ATV, approximate tumor volume; RTV, real tumor volume; TSAI, tumor surface area inner the intestine; TSAO, tumor surface area outside the intestine; TSA, total surface area of tumor; TC, tumor compactness.

preoperative CCRT, *sigmoid(X)* was the activation function of logistic regression. When the value of Z1/Z2 is greater than the corresponding optimal cutoff, the patient can be judged as insensitive to preoperative CCRT. In order to facilitate clinical application, we formulated

**Table 3. The variance inflation factors (VIFs) all variables in regression model.**

| Variables | Collinearity analysis | | | | | |
|---|---|---|---|---|---|---|
| | Including all variables | | Without ATV in variables | | Without RTV in variables | |
| | Tolerance | VIF | Tolerance | VIF | Tolerance | VIF |
| Age | 0.784 | 1.275 | 0.788 | 1.269 | 0.796 | 1.256 |
| Gender | 0.712 | 1.405 | 0.823 | 1.216 | 0.731 | 1.368 |
| Clinical T stage | 0.855 | 1.169 | 0.857 | 1.166 | 0.857 | 1.166 |
| Clinical N stage | 0.776 | 1.289 | 0.776 | 1.289 | 0.780 | 1.283 |
| pathological pattern | 0.887 | 1.127 | 0.890 | 1.124 | 0.890 | 1.123 |
| CEA | 0.775 | 1.290 | 0.796 | 1.257 | 0.804 | 1.244 |
| CD44v6 | 0530 | 1.887 | 0.553 | 1.807 | 0.549 | 1.821 |
| TML | 0.692 | 1.446 | 0.697 | 1.435 | 0.694 | 1.441 |
| TMD | 0.687 | 1.456 | 0.687 | 1.456 | 0.692 | 1.445 |
| ATV | 0.135 | 7.391 | | | 0.233 | 4.292 |
| RTV | 0.123 | 8.124 | 0.212 | 4.718 | | |
| TSAI | 0.827 | 1.210 | 0.833 | 1.200 | 0.845 | 1.184 |
| TSAO | 0.777 | 1.287 | 0.777 | 1.286 | 0.781 | 1.280 |
| TSA | 0.172 | 5.810 | 0.181 | 5.518 | 0.221 | 4.528 |
| TC | 0.695 | 1.438 | 0.726 | 1.378 | 0.699 | 1.430 |

CEA, Carcinoembryonic antigen; TML, tumor maximum longitudinal length; TMD, tumor maximum transverse diameter; ATV, approximate tumor volume; RTV, real tumor volume; TSAI, tumor surface area inner the intestine; TSAO, tumor surface area outside the intestine; TSA, total surface area of tumor; TC, tumor compactness.

**Table 4. Significantly predictors of no/low response to preoperative CCRT in LARC.**

| Significantly predictors | | Multivariate analysis | | | | | |
|---|---|---|---|---|---|---|---|
| | | B | S.E. | Exp(B) | Wald | P | 95% CI |
| Without ATV in model | | | | | | | |
| CD44v6 | X7 | 1.003 | 0.378 | 2.727 | 7.044 | 0.008 | 1.300–5.721 |
| RTV | X11 | 1.557 | 0.478 | 4.744 | 10.589 | 0.001 | 1.857–12.119 |
| TC | X15 | -1.044 | 0.440 | 0.352 | 5.630 | 0.018 | 0.149–0.834 |
| Constant | | -4.419 | 1.464 | 0.012 | 9.115 | 0.003 | |
| Without RTV in model | | | | | | | |
| CD44v6 | X7 | 0.864 | 0.327 | 2.373 | 6.973 | 0.008 | 1.250–4.506 |
| TSA | X14 | 2.081 | 0.754 | 8.010 | 7.618 | 0.006 | 1.828–35.103 |
| Constant | | -6.547 | 1.597 | 0.001 | 16.810 | 0.000 | |

ATV, approximate tumor volume; RTV, real tumor volume; TC, tumor compactness; TSA, total surface area of tumor.

nomogram (Figs 2 and 3) based on logistic regression analysis results, through which clinicians could evaluate the probability of no/low response of patients.

## Evaluation of predictive value

The results of predictive performance of above Logistic regression models and volumetric imaging parameters significantly associated with the no/low response to CCRT were obtained by ROC analysis (Figs 4 and 5). The Area Under Curve (AUC) of Z1, Z2, RTV, TC, and TSA were 0.900 (95%CI 0.811–0.965), 0.858(95%CI 0.761–0.926), 0.771(95%CI 0.663–0.858), 0.754 (95%CI 0.644–0.844), and 0.859 (95%CI 0.762–0.927), respectively. Based on the optimal cut-off values of 0.787, 0.658, 65.00 cm$^3$, 1.35, and 14.54cm$^2$, the sensitivity of Z1, Z2, RTV, TC, and TSA were 95.80%, 79.17%, 62.50%, 95.83%, 62.5%, the specificity were 70.90%, 74.55%, 83.64%, 47.27%, and 96.36%, the positive predictive values were 58.96%, 57.58%, 62.51%, 44.23%, and 88.23%, the negative predictive values were 97.48%, 89.13%, 83.64%, 96.29%, and 85.48%, respectively. These results suggest that the two regression models are of high predictive value, TC's predictive advantage is mainly reflected in its sensitivity, while TSA's is mainly reflected in its specificity.

## Discussion

As an important part of the comprehensive treatment for LARC, preoperative CCRT can provide better local control, toxicity profile, and sphincter preservation than postoperative CCRT [31, 32]. However, it usually increases the risk of pelvic edema or pelvic fibrosis, while increasing the difficulty of surgery. In addition, Bertucci et al. found that preoperative radiation was the single most significant and controllable risk factor predicting perineal wound failure [33]. Anastomotic leakage is a very serious complication after colorectal surgery, which was increased in patients having undergone preoperative CCRT [34, 35]. Therefore, in patients showing no/low response to neoadjuvant therapy, preoperative CCRT cannot only delay the timing of surgery and enhance the complexity and difficulty of surgery, but also increase the risk of the a forementioned postoperative complications.

Evidence suggests that Cancer stem cells (CSCs) are responsible for the growth and recurrence of tumors and their resistance to radiotherapy [36, 37]. The underlying mechanisms include that CSCs are usually in cells S/G0 phase [38], which have powerful functions of replication and DNA damage repair, while tumor cells during G2/M phases are the most sensitive to radiotherapy [39, 40]. Furthermore, radiation transforms the division strategy of CSCs from

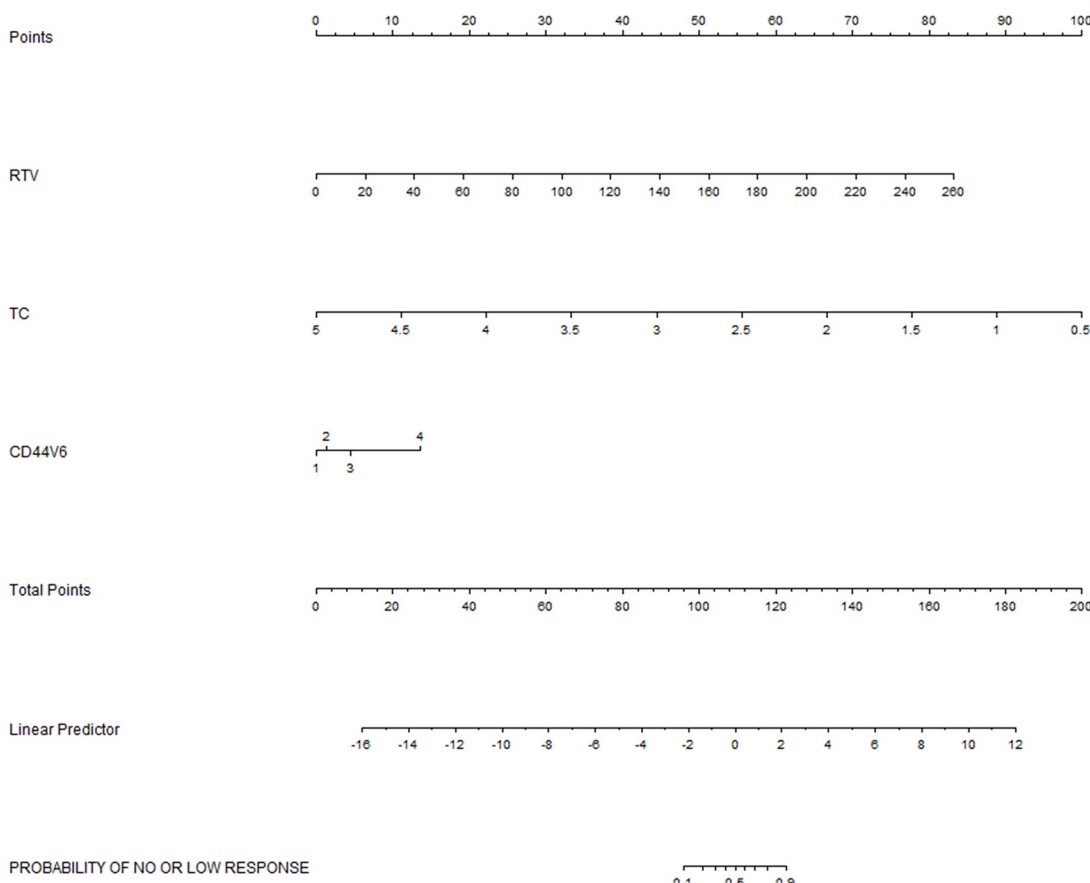

**Fig 2. Nomogram was used to predict the risk of no / low response of CCRT in LARC patients without ATV.**

asymmetry to symmetry, which in turn leads to an increase in the proportion of tumor stem cells either in proportion or in absolute numbers [41, 42]. It was found that CD44v6 is an important CSCs marker, its expression level was positively related to the resistance of chemotherapy and radiotherapy resistance of nasopharyngeal carcinoma, prostate cancer and rectal cancer [15, 16]. In this study, we found that the expression of CD44v6 was significantly higher in patients with chemoradiotherapy resistance than that of patients with chemoradiotherapy sensitivity and could be used as an independent predictor in order to predict the resistance of preoperative CCRT, which was consistent with the results of Huh et al. in screening for predictors of tumor regression after preoperative CCRT for rectal cancer [18].

Because the rectum is a hollow organ, the entire intestines at the tumor location are usually included in the scope of gross tumor volume (GTV) when the radiotherapy target is delineated. This will undoubtedly make the tumor volumetric parameters obtained greater than the true size of the tumor. RTV is the most precise parameter of the tumors volume obtained by eliminating intestinal volume. In this study, we analyzed correlations between the no/low response status of CCRT and RTV, ATV, respectively, and found that ATV could not be used as an independent predictor to predict the patient's resistance to CCRT, while RTV could. In previous studies, Chen et al. [43] found that GTV(actually ATV) was negatively correlated with the prognosis of patients with esophageal cancer after definitive radiotherapy, and numerous studies [22, 44, 45] on preoperative CCRT for rectal cancer have shown that tumor volume was negatively correlated with the pCR status after preoperative CCRT. Although the

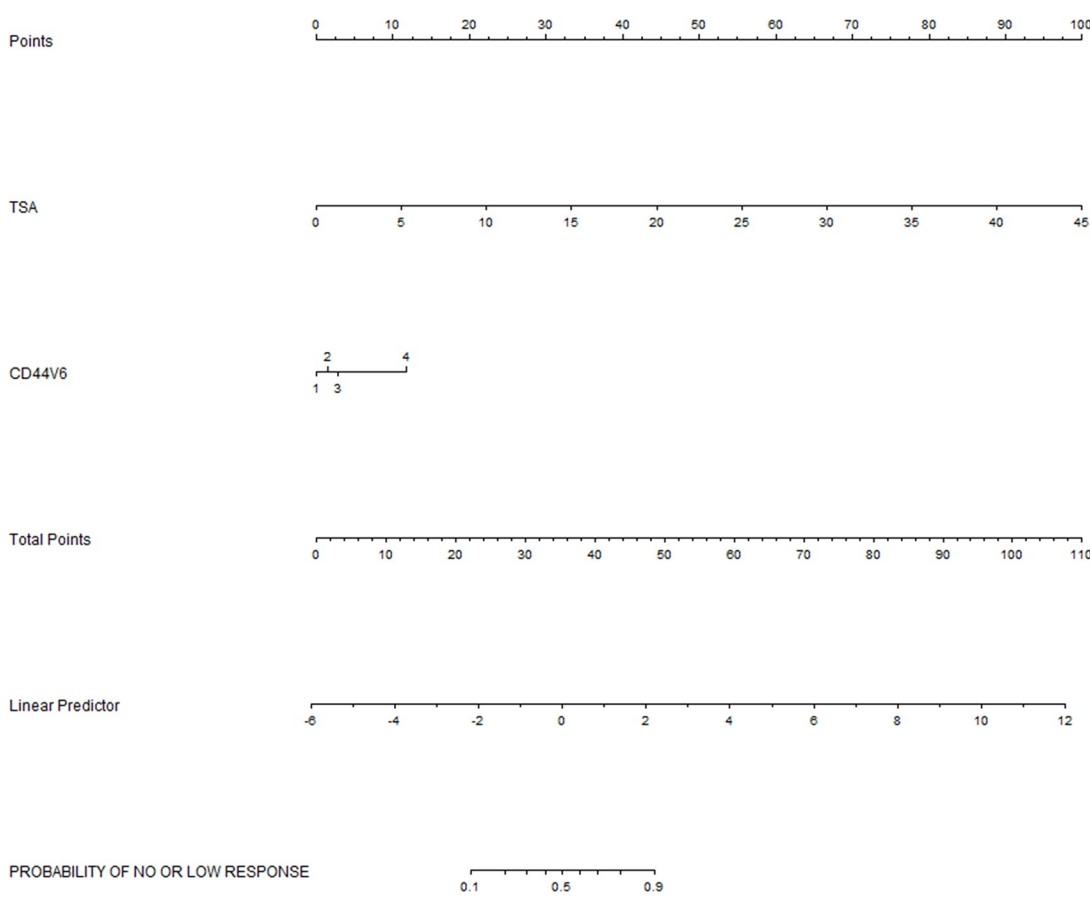

**Fig 3. Nomogram was used to predict the risk of no / low response of CCRT in LARC patients without RTV.**

endpoint of these studies were pCR, it also suggested to some extent that larger tumors may have more no/low response to CCRT; our results bear this out.

TC is a parameter originated from tumor-measurable volumetric imaging parameters and has been found to be closely related to tumor morphology and invasiveness [24, 46, 47]. According to Fave et al., TC not only reflects the information of tumor volume and TNM, but also the prognosis of non-small cell lung cancer patients [48]. From the equation of TC, there seems to be an inverse relationship between TC and TSA. It is well known that the value of TSA determines the degree of tumor contact with the surrounding tissues and organs. Studies by Agner et al. revealed that the TC value of triple-negative breast cancer was significantly higher than that of HER2-positive breast cancer, suggesting that the edge of triple-negative breast cancer is smoother than HER2-positive breast cancer [24]. Therefore, we hypothesized that the correlation between TC and tumor invasiveness may be due to the size of TSA. From the ROC curve analysis results, we found that the predictive specificity of TSA was 95.83%, while the predictive sensitivity of TC was 95.24%. This indicates that when the TSA value is greater than 14.54cm$^2$, 96.36% of patients with LARC, who have non-obvious resistance the preoperative CCRT, can be identified, and when the TC value is less than or equal to 1.35, 95.83% of patients with LARC, who show no/low response to preoperative CCRT, can be screened out.

In our study, two predictive models of no/low response status were obtained and their predictive values were analyzed by ROC. It was found that the AUC values of both models were

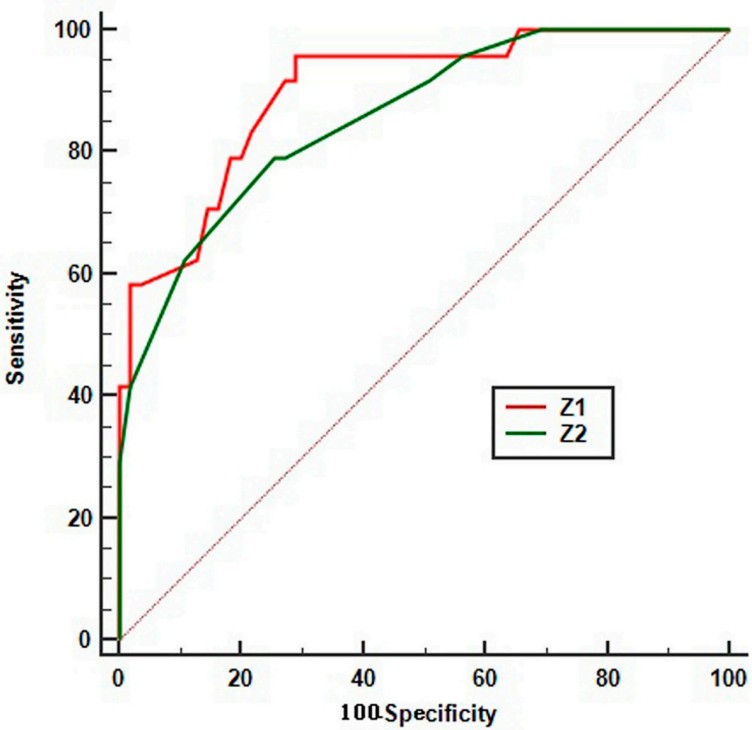

**Fig 4. ROC curve of logistic regression formulas using the no/low response status as test variable.** Z1 is the Logistic regression formula obtained after the deletion of ATV in the independent variable, Z2 is the Logistic regression formula obtained after the deletion of RTV in the independent variables.

greater than 0.85, which means that they had good predictive performance. Therefore, for initial LARC patients, we can put relevant parameters in place to predict the patient's CCRT resistance. When the Z value is greater than the corresponding cut-off value, it can be considered that the patient will show no/low response to preoperative CCRT. The application of these two models would be instructive for the selection of treatment strategies for LARC patients. In addition, we can also include the statistically significant predictor values, such as the expression levels of CD44v6, RTV, TC, and TSA, into the nomogram to assess the probability of no / low response to preoperative CCRT.

## Limitations

This study has some limitations. To begin with, only patients receiving long-course preoperative CCRT were included, and patients receiving short-course preoperative radiotherapy were not included, thus limiting the predictive application of the obtained regression model for these patients. What cannot be ignored, however, is the number of patients receiving long-course CCRT in both China and the United States, which is far more than those receiving short-course radiotherapy. Taking this into account, our results still have reasonable representativeness and will be followed-up to verify the results of this study in patients receiving short-course preoperative radiotherapy. Another limitation of this study was that the pathology of pelvic lymph nodes could not be accurately obtained before surgery, therefore, imaging information of suspicious positive lymph nodes found in MRIs were not included as imaging parameters in the study, which will undoubtedly lead to the limitations of our results. The third limitation of this study is that the results we obtained have not appeared in the previous

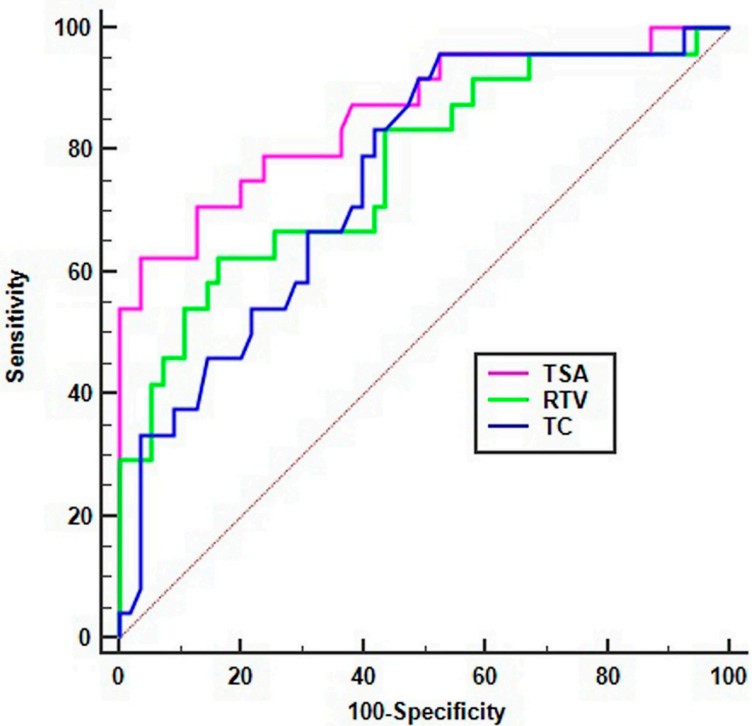

**Fig 5. ROC curve of RTV, TC and TSA using the no/low response status as test variable.** RTV represents real tumor volume, TSA represents total surface area of tumor, TC represents tumor compactness.

reports, plus there are fewer patients in this study (79 cases), so a larger cohort study still needs to be carried out. In addition, the practicability of our results also needs to be further verified in clinical practice. The final limitation was that we did not include the detection index of immunotherapy in our candidate factors. At present, studies have found that there is a correlation between the status of immunoassay indicators and the response of LARC patients to CCRT. For example, Hasan et al. analyzed the relationship between the response of 5086 rectal cancer patients to preoperative CCRT and microsatellite stability (MSI) by using the data of National Cancer Database (NCDB), and found that MSI(+) was related to the low pCR rate after CCRT [49]. Since Hasan's study was published in 2018 and our experiment started in 2015, this study did not include immunoassay indexes such as PD-1, PDL-1, MSI, and tumor cumulative burden (TMB). Despite such limitations, this study offers a unique perspective for the choice of treatment options for LARC patients.

## Conclusion

The logistic regression model we obtained is a method for preoperative prediction of patients' suitability for neoadjuvant chemoradiotherapy. The application of this method would reduce the risk probability of those patients, who are not sensitive to CCRT, from suffering the corresponding radiation complications as well as longer waiting periods for surgery and would also provide evidence for individualized treatment options for LARC patients. Independent predictors for resistance status include CD44v6 expression level and tumor imaging volumetric parameters such as real tumor volume (RTV), tumor surface area (TSA) and tumor compactness (TC). These factors are of great value in predicting no/low response status of LARC patients to preoperative CCRT.

## Supporting information

**S1 Data. Data of influencing factors.**
(XLSX)

## Acknowledgments

We would like to thank Yue ping. Liu, Hui chai. Yang, Zi feng. Chi and Ruo hui. Zhang for their technical assistance.

## Author Contributions

**Conceptualization:** Fengpeng Wu.

**Data curation:** Fengpeng Wu, Guiying Wang.

**Funding acquisition:** Fengpeng Wu, Guiying Wang.

**Investigation:** Fengpeng Wu, Guiying Wang, Congrong Yang.

**Methodology:** Fengpeng Wu, Guiying Wang, Jun Wang, Chaoxi Zhou, Congrong Yang.

**Project administration:** Fengpeng Wu, Guiying Wang, Wenbo Niu.

**Resources:** Fengpeng Wu, Guiying Wang, Jun Wang, Chaoxi Zhou, Wenbo Niu, Jianfeng Zhang, Guanglin Wang, Yafan Yang.

**Supervision:** Guiying Wang.

**Validation:** Chaoxi Zhou.

**Writing – original draft:** Fengpeng Wu, Guiying Wang.

**Writing – review & editing:** Fengpeng Wu, Guiying Wang.

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
