## [Decision Letter · Decision Letter 0]

31 Dec 2019

PONE-D-19-31144

Analysis of influencing factors of no/low response to preoperative concurrent chemoradiotherapy in locally advanced rectal cancer

PLOS ONE

Dear Dr. Wang,

Thank you for submitting your manuscript to PLOS ONE. After careful consideration, we feel that it has merit but does not fully meet PLOS ONE’s publication criteria as it currently stands. Therefore, we invite you to submit a revised version of the manuscript that addresses the points raised during the review process.

ACADEMIC EDITOR: 

1. In the abstract, "15 factors" is better to be written as "Fifteen factors..."

2. In the letter to the editor, "myco-authors" should be "my co-authors."

3. Line 68, "CD44v6," should not be Italic here. It is supposed to be a protein stained by an IHC method.  Please make sure all the CD44v6 in the manuscript has been correctly expressed. 

4. Line 69-70, "Breast cancer specimens were used to act as positive control samples, PBS (pH 7.4) was used as a negative control"  They should be two different sentences. 

5. The authors said that they used breast cancer samples for control. Was the breast cancer sample prospectively taken or a control slide? Otherwise, please make a clear statement of how the authors used breast cancer samples within the approval range of the IRB. 

5. All the tables lack abbreviations independently. Please add them in the bottom of each tables.  

6. No potential limitations were stated in the manuscript. Please kindly consider providing one paragraph for the limitations of the present study.

7. In conclusion, the authors are recommended to be more humble. The findings here have not been independently validated. Please do not conclude that some patients should be "saved"  from the corresponding radiation complications. The conclusion is too aggressive and not scientific enough.==============================

We would appreciate receiving your revised manuscript by Feb 14 2020 11:59PM. To enhance the reproducibility of your results, we recommend that if applicable you deposit your laboratory protocols in protocols.io, where a protocol can be assigned its own identifier (DOI) such that it can be cited independently in the future. For instructions see: http://journals.plos.org/plosone/s/submission-guidelines#loc-laboratory-protocols

We look forward to receiving your revised manuscript.

Kind regards,

Jason Chia-Hsun Hsieh, M.D. Ph.D

Academic Editor

PLOS ONE

Journal Requirements:

2. We noticed you have some minor occurrence(s) of overlapping text with the following previous publication(s), which needs to be addressed:

https://doi.org/10.1093/jrr/rrz035

In your revision ensure you cite all your sources (including your own works), and quote or rephrase any duplicated text outside the Methods section. Further consideration is dependent on these concerns being addressed.

Additional Editor Comments (if provided):

1. In the abstract, "15 factors" is better to be written as "Fifteen factors..."

2. In the letter to the editor, "myco-authors" should be "my co-authors."

3. Line 68, "CD44v6," should not be Italic here. It is supposed to be a protein stained by an IHC method. Please make sure all the CD44v6 in the manuscript has been correctly expressed.

4. Line 69-70, "Breast cancer specimens were used to act as positive control samples, PBS (pH 7.4) was used as a negative control" They should be two different sentences.

5. The authors said that they used breast cancer samples for control. Was the breast cancer sample prospectively taken or a control slide? Otherwise, please make a clear statement of how the authors used breast cancer samples within the approval range of the IRB.

5. All the tables lack abbreviations independently. Please add them in the bottom of each tables.

6. No potential limitations were stated in the manuscript. Please kindly consider providing one paragraph for the limitations of the present study.

7. In conclusion, the authors are recommended to be more humble. The findings here have not been independently validated. Please do not conclude that some patients should be "saved" from the corresponding radiation complications. The conclusion is too aggressive and not scientific enough.

Reviewers' comments:

Reviewer's Responses to Questions

**Comments to the Author**

1. Is the manuscript technically sound, and do the data support the conclusions?

Reviewer #1: No

Reviewer #2: Yes

2. Has the statistical analysis been performed appropriately and rigorously? 

Reviewer #1: Yes

Reviewer #2: I Don't Know

3. Have the authors made all data underlying the findings in their manuscript fully available?

Reviewer #1: Yes

Reviewer #2: Yes

4. Is the manuscript presented in an intelligible fashion and written in standard English?

Reviewer #1: No

Reviewer #2: No

5. Review Comments to the Author

Reviewer #1: The study included 79 patients with locally advanced rectal cancer with the inferior margin of tumor within 10 cm from the anal verge, clinical T2N+ or clinical T3-4 who were qualified for a long course preoperative CCRT. After a long course of preoperative CCRT all the patients were qualified for TME surgery. 15 parameters were evaluated to predict the low/no response to preoperative CCRT. I have a several doubts according to the qualification for CCRT, the type of surgery which was performed, the structure of the study, discussion and conclusions:

1)the patients qualified for the therapy had upper rectal cancer (10 cm from the anal verge)-from the data we do not know, which cancers were above peritoneal reflection (>12 cm from the anal verge)these do not benefit from CRT and should be treated as colon cancer

2)in discussion there are mentioned disadvantages of CRTthe problems with peritoneal wound healing (none of patients had peritoneal woundall had TME, no abdominoperineal resections), anastomotic leakage (no data about it, we do not know if the patients had protective artificial anus)

3)the factors which are the most important during surgery and in the local treatment of the rectal cancer are clear margins (circumferentialpositive in the group with low/no response to preoperative CCRT5 patients, negative in the group with obvious response to preoperative CCRT0 patients, the authors do not mention proximal and distal margins which are also important). We do not have data about MRI images after preoperative CCRT before surgeryif the clear margins in advanced cT3-T4 are at risk (possible R1 or R2 resection), why to perform TMEthese patients should undergo multivisceral R0 resection if it is technically possible; we do not have information in which cT the margins were positive

4) not all cT2N+/T3 tumors need preoperative CCRT if the quality of TME is highwe do not have any information about the quality of surgery

5)even the low response to the preoperative CCRT may mean R0 surgerythe shrinkage of the tumor should result in a clear CRM (>1mm) so we should not discourage these patients from the preoperative CCRT; the pathologic regression in 24 patients meant no/low response, but only in 5 patients the CRM was positive

6)emotional, general expressions like "in the real world" (line 34) "a catastrophe for the patients" (line 174) should be omitted

7)endorectal ultrasound is not used for diagnosis of distant metastases (line 53)

7)grammatical and structural mistakes (line 40, 47, 48, 52, 164, 165, 179, 188, 203)

Reviewer #2: This study by Wang et al. tries to identify independent prognostic factors of poor response of CRT in rectal cancer.

Here are my main points which could help to improve the manuscript.

1. Table 1 should be replaced in results

2. English by place could be improved by a native speaker

3. The authors should better explain in Introduction what is CD44v6 and why it could be relevant in rectal cancer irradiation.

4. It remains unclear whether poor response was defined according to pathological examination or with preoperative MRI.

5. The authors report important results which could help decision making in the future. However, I suggest that an external cohort could validate these results.

6. Again, to improve the manuscript, a prognostic nomogram including all relevant parameters should be proposed and would greatly improve the quality of the manuscript.

7. Possible bias should be discussed.

6. PLOS authors have the option to publish the peer review history of their article (what does this mean?). If published, this will include your full peer review and any attached files.

Reviewer #1: No

Reviewer #2: No

---

## [Author Response · Author response to Decision Letter 0]

20 Feb 2020

Dear professor:

Thank you very much for your suggestion and guidance on our manuscript. After consulting the relevant literatures, we have revised our paper according to your suggestions and questions. Our answer as follows: 

Response to Editor:

Our answer to question one ( In the abstract, "15 factors" is better to be written as "Fifteen factors...") is as follows:

Thank you very much for your advice. In the abstract section, we have modified this error.

Our answer to question two (In the letter to the editor, "myco-authors" should be "my co-authors.") is as follows:

Thank you very much for your reminding. In the letter to the editor, we have modified this error.

Our answer to question three (Line 68, "CD44v6," should not be Italic here. It is supposed to be a protein stained by an IHC method. Please make sure all the CD44v6 in the manuscript has been correctly expressed. ) is as follows:

Thank you very much for your reminding. We carefully checked the entire manuscript and made comprehensive corrections to the writing errors of CD44v6. Thanks again for your advice.

Our answer to question four (Line 69-70, "Breast cancer specimens were used to act as positive control samples, PBS (pH 7.4) was used as a negative control" They should be two different sentences. ) is as follows:

Thank you very much for your advice. We have modified this error accordingly in our revised version.

Our answer to question five (The authors said that they used breast cancer samples for control. Was the breast cancer sample prospectively taken or a control slide? Otherwise, please make a clear statement of how the authors used breast cancer samples within the approval range of the IRB. ) is as follows:

Thank you very much for your advice. Due to our negligence, the reason for using breast cancer samples as a positive control is not described in the manuscript. In this revised manuscript, we have shown the corresponding basis. Thank you again for your reminding.

Our answer to question six (All the tables lack abbreviations independently. Please add them in the bottom of each tables.) is as follows:

Thank you very much for your reminding. We have added abbreviations in the bottom of each table in revised version.

Our answer to question seven (No potential limitations were stated in the manuscript. Please kindly consider providing one paragraph for the limitations of the present study.) is as follows:

Thank you very much for your reminding. With your suggestion, we carefully read relevant literatures and conducted in-depth thinking and discussion. We found that our study does have some limitations, so we elaborated them one by one in the discussion section of the revised version. 

Our answer to question eight (In conclusion, the authors are recommended to be more humble. The findings here have not been independently validated. Please do not conclude that some patients should be "saved" from the corresponding radiation complications. The conclusion is too aggressive and not scientific enough.) is as follows:

Thank you very much for your advice. Your criticism is correct and we have accepted it sincerely. In the conclusion section, we have deleted and modified those inappropriate statements. 

Response to Reviewer#1:

Our answer to question one (The patients qualified for the therapy had upper rectal cancer (10 cm from the anal verge)-from the data we do not know, which cancers were above peritoneal reflection (>12 cm from the anal verge)these do not benefit from CRT and should be treated as colon cancer) is as follows:

Thank you very much for your reminding. With your suggestion, we carefully read relevant literatures. The relevant contents we have read are reported as follows: The rectum is defined as the distal 16 cm of the large intestine leading to the anal canal, and a cancer between 12 and 16 cm from the anal verge is classified as a rectal cancer of the upper third[1,2]. Previous studies have shown that neoadjuvant chemoradiotherapy have no significant effect on the upper rectal cancer [3, 4]. At present, most scholars believe that preoperative CCRT is necessary for tumor under the anterior peritoneal reflection. Memon 's study showed that the mean distance from the anal verge to the anterior peritoneal reflection was 11.9 cm (men) and 10 cm (women) [5]. Studies by Peeters found that radiotherapy has a significant effect on reducing the risk of local recurrence in patients with lesions between 5 and 10 cm from the anal verge [4]. In China, most scholars defined between 10 and 15 cm from the anal verge as the upper rectum, 5-10 cm from the anal verge as the middle rectum, and 5 cm above the anal verge as the lower rectum [6]. Therefore, in this study, we took the distance of less than 10cm between the inferior margin of the tumor and the anal verge less than 10cm as one of the inclusion criteria.

References

1. Soreide O, Norstein J, Fielding LP, et al. International standardization and documentation of the treatment of rectal cancer. In:Soreide O, Norstein J, eds. Rectal Cancer Surgery. Optimization-Standardization-Documentation. Berlin, Germany: Springer;1997:405– 445. 

2. Wittekind C, Greene FL, Henson C, eds. UICC. TNM Classification of Malignant Tumors, 6th ed. New York, NY: John Wiley & Sons; 2002.

3. Kapiteijn E, Marijnen C, Nagtegaal ID, et al. Preoperative radiotherapy combined with total mesorectal excision for resectable rectal cancer. N Engl J Med. 2001;345:638–646.

4. Peeters KCMJ, Marijnen CAM, Nagtegaal ID, et al. The TME trial after a median follow-up of 6 years: increased local control but no survival benefit in irradiated patients with resectable rectal carcinoma. Ann Surg. 2007;246:693–701.

5. Memon S, Keating JP, Cooke HS, et al. A study into external rectal anatomy: improving patient selection for radiotherapy for rectal cancer. Dis Colon Rectum. 2009 Jan; 52(1):87-90. 

6. Wen xin, LIU Lei , XIA Jin dong, et al. MRI in determining the location of rectal cancer with respect to the peritoneal reflection. J Surg Concepts Pract (china) 2013:18(5):446-449.

Our answer to question two (in discussion there are mentioned disadvantages of CRTthe problems with peritoneal wound healing (none of patients had peritoneal woundall had TME, no abdominoperineal resections), anastomotic leakage (no data about it, we do not know if the patients had protective artificial anus)) is as follows:

Thank you for your advice. We have supplemented this part in the "Operation and Pathologic regression status after CCRT" section of the revised version, as described next: “Out of all enrolled patients, 17 underwent Miles surgery, 62 underwent Dixon surgery, none of whom underwent Hartmann surgery. Of the 62 patients who underwent Dixon surgery, 12 underwent a prophylactic terminal ileostomy”. Thank you again for your reminding.

Our answer to question three (the factors which are the most important during surgery and in the local treatment of the rectal cancer are clear margins (circumferentialpositive in the group with low/no response to preoperative CCRT5 patients, negative in the group with obvious response to preoperative CCRT0 patients, the authors do not mention proximal and distal margins which are also important). We do not have data about MRI images after preoperative CCRT before surgeryif the clear margins in advanced cT3-T4 are at risk (possible R1 or R2 resection), why to perform TMEthese patients should undergo multivisceral R0 resection if it is technically possible; we do not have information in which cT the margins were positive) is as follows:

Thank you for your reminding. Due to negligence, we did not show the situation of the proximal and distal margins in the manuscript, which caused you a lot of trouble in review, and we apologize deeply. We have supplemented this result in the "Operation and Pathologic regression status after CCRT" section of the revised version, as described next: “The proximal and distal margins of all specimens were negative.” 

In addition, the MRI images (after preoperative CCRT before surgery) showed that the status of CRM of all enrolled patients was negative, while the postoperative pathology showed that the CRM of 5 patients were positive, indicating that there was an error in the preoperative MRI evaluation of CRM. Therefore, we did not show them in the original manuscript. After receiving your suggestion, we realized the importance of presenting the CRM status of MRIs, so we supplemented these information in "Table 1: baseline demographics of all patients" in the revised version. Thank you again for your reminding.

Our answer to question four (not all cT2N+/T3 tumors need preoperative CCRT if the quality of TME is highwe do not have any information about the quality of surgery) is as follows:

Thank you for your advice. At present, preoperative CCRT combined with TME as the standard treatment for LARC patients has been written into many guidelines. In clinical practice, as you said, we did experience many patients with cT2N+/T3 who did not undergo preoperative CCRT but preferred TME surgery and achieved good results. However, 79 patients enrolled in this study were given the treatment plan at the experimental design stage, so they all adopted preoperative CCRT. With your suggestion, we supplemented the information about the quality of surgery in the" Preoperative CCRT and surgery" section of the revised manuscript as follows: “Total mesorectal excision (TME) surgery was performed by two surgeons with more than 15 years of experience at 8 to 10 weeks after CCRT, each of whom completed at least 100 rectal cancer surgeries per- year”. Thank you again for your reminding.

Our answer to question five (even the low response to the preoperative CCRT may mean R0 surgerythe shrinkage of the tumor should result in a clear CRM (>1mm) so we should not discourage these patients from the preoperative CCRT; the pathologic regression in 24 patients meant no/low response, but only in 5 patients the CRM was positive) is as follows:

What you said is very reasonable. It is true that many patients did not reach the stage of pathological decline after receiving the neoadjuvant therapy, but the tumor volume was reduced, thus R0 resection was performed. What cannot be ignored, however, is that there are still some patients not only did not achieve pathological decline and tumor volume shrinkage after CCRT, but also developed further aggravation. We all know that perioperative adjuvant treatment includes preoperative treatment and postoperative treatment. If we can identify these patients who are not sensitive to preoperative CCRT before treatment (the purpose of this study), give them active surgical treatment and then carry out postoperative radiotherapy and systemic treatment, these patients may receive greater benefits.

Our answer to question six (emotional, general expressions like "in the real world" (line 34) "a catastrophe for the patients" (line 174) should be omitted) is as follows:

Thank you for your advice. We have deleted these contents in the revised manuscript.

Our answer to question seven (endorectal ultrasound is not used for diagnosis of distant metastases (line 53)) is as follows:

Thank you for your advice. We have deleted this content in the revised manuscript. Thank you again for your reminding.

Our answer to question eight (grammatical and structural mistakes (line 40, 47, 48, 52, 164, 165, 179, 188, 203)) is as follows:

Thank you for your advice. With your suggestion, we invited native English speakers to proofread the language of this manuscript.

Response to Reviewer#2:

Our answer to question one (Table 1 should be replaced in results) is as follows:

According to your opinion, we rearranged the pathological results related to tumor regression in Table 1 into the results section of this manuscript.

Our answer to question two (English by place could be improved by a native speaker) is as follows:

Thank you for your advice. With your suggestion, we invited native English speakers to proofread the language of this manuscript.

Our answer to question three (The authors should better explain in Introduction what is CD44v6 and why it could be relevant in rectal cancer irradiation) is as follows:

Your opinion is reasonable. We have explained CD44v6 in the introduction section of the revised version. Thank you for your advice.

Our answer to question four (It remains unclear whether poor response was defined according to pathological examination or with preoperative MRI) is as follows:

Thank you for your reminding. We know that as you said, there are no unified standards for evaluating the efficacy of preoperative CCRT in patients with rectal cancer. Common methods include preoperative imaging evaluation (especially MRI) and postoperative pathology evaluation. However, these methods focus more on patients who are sensitive to neoadjuvant therapy because some of these patients may choose the "watch and wait" strategy. But even among those patients who were evaluated as cCR, some relapsed during the "watch and wait" strategy. In clinical practice, we often meet some patients who underwent MRI examination 8-10 weeks after receiving the preoperative CCRT and found that the imaging T stage did not change significantly from the original MRI results, but the postoperative pathology displayed tumor regression. Therefore, the tumor regression status indicated by MRI after CCRT and before surgery can not be used as the gold standard for evaluating the response of lesions to CCRT, but can only be used as an important reference.

Our answer to question five (The authors report important results which could help decision making in the future. However, I suggest that an external cohort could validate these results) is as follows:

Thank you for your reminding. Your suggestion is very important for our follow-up work. In the follow-up work, we will not only verify it in patients with long-course preoperative CCRT, but also in patients with short-course preoperative radiotherapy. These contents are described in the Limitations section of the revised version.

Our answer to question six (Again, to improve the manuscript, a prognostic nomogram including all relevant parameters should be proposed and would greatly improve the quality of the manuscript.) is as follows:

Thank you very much for your reminding. We have drawn the nomogram chart in the revised draft and elaborated it in the results and discussion section. Thanks again for your advice.

Our answer to question seven (Possible bias should be discussed) is as follows:

According to your opinion, I discussed the bias and limitations of our study in the Discussion section of the revised version.

Thank you and best regards.

Yours sincerely,

The author of PONE-D-19-31144

February 20, 2020

---

## [Decision Letter · Decision Letter 1]

30 Mar 2020

PONE-D-19-31144R1

Analysis of influencing factors of no/low response to preoperative concurrent chemoradiotherapy in locally advanced rectal cancer

PLOS ONE

Dear Dr Wang,

Thank you for submitting your manuscript to PLOS ONE. After careful consideration, we feel that it has merit but does not fully meet PLOS ONE’s publication criteria as it currently stands. Therefore, we invite you to submit a revised version of the manuscript that addresses the points raised during the review process.

ACADEMIC EDITOR: The manuscript has been improved. However, some crucial issues require revision. 

1. Please respond to the reviewers' comments in a point-by-point manner.  

2. The authors wrote that "the tumor stem cell marker CD44v6, which has been reported to be related to the resistance of neoadjuvant chemoradiation in rectal cancer [5]." The writing was not clear for readers who are not working in this field. Please add one paragraph (or several sentences) to briefly introduce the role of CD44v6 in colorectal cancer and describe the reason(s) why the study chose this factor. 

We would appreciate receiving your revised manuscript by May 14 2020 11:59PM. To enhance the reproducibility of your results, we recommend that if applicable you deposit your laboratory protocols in protocols.io, where a protocol can be assigned its own identifier (DOI) such that it can be cited independently in the future. For instructions see: http://journals.plos.org/plosone/s/submission-guidelines#loc-laboratory-protocols

We look forward to receiving your revised manuscript.

Kind regards,

Jason Chia-Hsun Hsieh, M.D. Ph.D

Academic Editor

PLOS ONE

Additional Editor Comments (if provided):

The manuscript has been improved. However, some crucial issues require revision.

1. Please respond to the reviewers' comments in a point-by-point manner.

2. The authors wrote that "the tumor stem cell marker CD44v6, which has been reported to be related to the resistance of neoadjuvant chemoradiation in rectal cancer [5]." The writing was not clear for readers who are not working in this field. Please add one paragraph (or several sentences) to briefly introduce the role of CD44v6 in colorectal cancer and describe the reason(s) why the study chose this factor.

Reviewers' comments:

Reviewer's Responses to Questions

**Comments to the Author**

1. If the authors have adequately addressed your comments raised in a previous round of review and you feel that this manuscript is now acceptable for publication, you may indicate that here to bypass the “Comments to the Author” section, enter your conflict of interest statement in the “Confidential to Editor” section, and submit your "Accept" recommendation.

Reviewer #1: All comments have been addressed

Reviewer #2: (No Response)

2. Is the manuscript technically sound, and do the data support the conclusions?

Reviewer #1: Yes

Reviewer #2: Partly

3. Has the statistical analysis been performed appropriately and rigorously? 

Reviewer #1: Yes

Reviewer #2: N/A

4. Have the authors made all data underlying the findings in their manuscript fully available?

Reviewer #1: Yes

Reviewer #2: Yes

5. Is the manuscript presented in an intelligible fashion and written in standard English?

Reviewer #1: Yes

Reviewer #2: No

6. Review Comments to the Author

Reviewer #1: I have several doubts and proposals for improvement of the article:

1)line 36"preoperative CCRT will result"I would change into" preoperative CCRT may result"

2)line 51 should be August 2017

3)line 117-120. I have doubts about terminology of surgical procedures and surgical technique: Miles technique=abdominoperineal resection is always associated with end-colostomy formation,

Dixon surgery-a type of blunt dissection"Until the late 1970s, anterior resection with blunt dissection of the mid and distal rectum (as described by Dixon) continued to have a disease-free five-year survival rate for all stages treated with curative intent not exceeding 50% with a local recurrence rate of up to 20%. This was mainly related to the breaches often created on the mesorectal fascia and the mesorectum itself during the blunt rectal dissection". Nowadays a standard procedure is TME introduced be RJ Heald-"A surgical plane is a “potential space between contiguous organs which can be reproducibly created by dissection” . In rectal surgery the plane develops between the mesorectum and the surrounding somatic structures . Dissection along this plane should be sharp, under direct vision and gentle continuous traction".

Question about ileostomyusually the temporary protective loop ileostomy is performed, not a terminal ileostomy.

Article about the terminology and technique-

Minim Invasive Ther Allied Technol 25 (5), 226-33 Oct 2016

Techniques and Technology Evolution of Rectal Cancer Surgery: A History of More Than a Hundred Years

Marco Maria Lirici , Cristiano G S Hüscher

Reviewer #2: 1. Table 1 has not been moved to the Results Scetion

2. The role of CD44v6 is still little described in Introduction

3. Response to my fourth question remains unclear. Please provide a clear respsonse and define in Methods whether poor response was defined according to preoperative MRI or to pathological examination.

4. Response to my fifth question is again approximate. I suggest to add to Limitations that an external cohort could be useful to validate this results and would strengthen your results.

5. Figure 2 is unreadable.

7. PLOS authors have the option to publish the peer review history of their article (what does this mean?). If published, this will include your full peer review and any attached files.

Reviewer #1: Yes: Maciej Sebastian

Reviewer #2: No

---

## [Author Response · Author response to Decision Letter 1]

21 Apr 2020

Dear professor:

Thank you very much for your suggestion and guidance on our manuscript. After consulting the relevant literatures, we have revised our paper according to your suggestions and questions. Our answer as follows: 

Response to Editor:

Our answer to question one (Please respond to the reviewers' comments in a point-by-point manner) is as follows:

Thank you very much for your advice. In this reply, we give a point-to-point response to each comment of the reviewers.

Our answer to question two (The authors wrote that "the tumor stem cell marker CD44v6, which has been reported to be related to the resistance of neoadjuvant chemoradiation in rectal cancer [5]." The writing was not clear for readers who are not working in this field. Please add one paragraph (or several sentences) to briefly introduce the role of CD44v6 in colorectal cancer and describe the reason(s) why the study chose this factor.) is as follows:

Thank you very much for your reminding. In the introduction of this revised manuscript, we added a paragraph to describe the relationship between the expression of CD44v6 and the patient's response to radiation.

Response to Reviewer#1:

Our answer to question one (line 36"preoperative CCRT will result"I would change into" preoperative CCRT may result") is as follows:

Thank you very much for your reminding. We corrected this error in the revised manuscript.

Our answer to question two (line 51 should be August 2017) is as follows:

Thank you for your advice. We corrected this error in the revised manuscript.

Our answer to question three (line 117-120. I have doubts about terminology of surgical procedures and surgical technique: Miles technique=abdominoperineal resection is always associated with end-colostomy formation,Dixon surgery-a type of blunt dissection"Until the late 1970s, anterior resection with blunt dissection of the mid and distal rectum (as described by Dixon) continued to have a disease-free five-year survival rate for all stages treated with curative intent not exceeding 50% with a local recurrence rate of up to 20%. This was mainly related to the breaches often created on the mesorectal fascia and the mesorectum itself during the blunt rectal dissection". Nowadays a standard procedure is TME introduced be RJ Heald-"A surgical plane is a “potential space between contiguous organs which can be reproducibly created by dissection” . In rectal surgery the plane develops between the mesorectum and the surrounding somatic structures . Dissection along this plane should be sharp, under direct vision and gentle continuous traction".

Question about ileostomyusually the temporary protective loop ileostomy is performed, not a terminal ileostomy.

Article about the terminology and technique-Minim Invasive Ther Allied Technol 25 (5), 226-33 Oct 2016 Techniques and Technology Evolution of Rectal Cancer Surgery: A History of More Than a Hundred Years Marco Maria Lirici , Cristiano G S Hüscher) is as follows: 

Thank you very much for your reminding. After carefully reading the article you recommended, we modified the non-standard description in our article. At present, TME operation has been widely used in mainland China, but most clinicians still use the original operation name in their daily work, for example, the APR operation is called Miles, and the low anterior resection is called Dixon. This habitual title is not rigorous. Your reminder is very meaningful to us. We will correct it in our future clinical work and scientific research writing. In addition, in our revised version, terminal ileostomy was changed to temporary protective loop ileostomy. Thanks again!

Response to Reviewer#2:

Our answer to question one (Table 1 has not been moved to the Results Scetion) is as follows:

According to your opinion, we moved Table 1 to the result section and briefly described it in the revised manuscript.

Our answer to question two (The role of CD44v6 is still little described in Introduction) is as follows:

Thank you for your advice. In the introduction of this revised manuscript, we added a paragraph to describe the relationship between the expression of CD44v6 and the patient's response to radiation.

Our answer to question three (Response to my fourth question remains unclear. Please provide a clear respsonse and define in Methods whether poor response was defined according to preoperative MRI or to pathological examination.) is as follows:

Your opinion is reasonable. We added this part in the materials and methods of this revised manuscript, and described the relevant results in our results section.

Our answer to question four (Response to my fifth question is again approximate. I suggest to add to Limitations that an external cohort could be useful to validate this results and would strengthen your results.) is as follows:

Thank you for your reminding. We added this part to the Limitations of this revised manuscript. The details are as follows: The third limitation of this study is that the results we obtained have not appeared in the previous reports, plus there are fewer patients in this study (79 cases), so a larger cohort study still needs to be carried out. In addition, the practicability of our results also needs to be further verified in clinical practice.

Our answer to question five (Figure 2 is unreadable.) is as follows:

Thank you for your reminding. In this picture upload, we divided figure 2 into Fig.2 and Fig.3.

Thank you and best regards.

Yours sincerely,

The author of PONE-D-19-31144

April 21, 2020

---

## [Decision Letter · Decision Letter 2]

26 May 2020

Analysis of influencing factors of no/low response to preoperative concurrent chemoradiotherapy in locally advanced rectal cancer

PONE-D-19-31144R2

Dear Dr. Wang,

We are pleased to inform you that your manuscript has been judged scientifically suitable for publication and will be formally accepted for publication once it complies with all outstanding technical requirements.

With kind regards,

Jason Chia-Hsun Hsieh, M.D. Ph.D

Academic Editor

PLOS ONE

Additional Editor Comments (optional):

All the issues were addressed adequately.

Reviewers' comments:

Reviewer's Responses to Questions

**Comments to the Author**

1. If the authors have adequately addressed your comments raised in a previous round of review and you feel that this manuscript is now acceptable for publication, you may indicate that here to bypass the “Comments to the Author” section, enter your conflict of interest statement in the “Confidential to Editor” section, and submit your "Accept" recommendation.

Reviewer #1: All comments have been addressed

Reviewer #2: All comments have been addressed

2. Is the manuscript technically sound, and do the data support the conclusions?

Reviewer #1: Yes

Reviewer #2: Yes

3. Has the statistical analysis been performed appropriately and rigorously? 

Reviewer #1: Yes

Reviewer #2: Yes

4. Have the authors made all data underlying the findings in their manuscript fully available?

Reviewer #1: Yes

Reviewer #2: Yes

5. Is the manuscript presented in an intelligible fashion and written in standard English?

Reviewer #1: Yes

Reviewer #2: Yes

6. Review Comments to the Author

Reviewer #1: After all the changes which were made by the authors article is now well readable, technically sound and correct and leads to the correct conclusions.

I would only correct two vocabulary mistakes:

line 34-should be "clinical practice"

line 281 and 283-should be "LARC"

Reviewer #2: (No Response)

7. PLOS authors have the option to publish the peer review history of their article (what does this mean?). If published, this will include your full peer review and any attached files.

Reviewer #1: Yes: Maciej Sebastian

Reviewer #2: No

---

## [Editor Report · Acceptance letter]

29 May 2020

PONE-D-19-31144R2 

Analysis of influencing factors of no/low response to preoperative concurrent chemoradiotherapy in locally advanced rectal cancer 

Dear Dr. Wang:

I am pleased to inform you that your manuscript has been deemed suitable for publication in PLOS ONE. Congratulations! Your manuscript is now with our production department. 

With kind regards,

on behalf of

Dr. Jason Chia-Hsun Hsieh 

Academic Editor

PLOS ONE